# Molecular Detection of Non-O157 Shiga Toxin-Producing *Escherichia coli* (STEC) Directly from Stool Using Multiplex qPCR Assays

**DOI:** 10.3390/microorganisms10020329

**Published:** 2022-01-31

**Authors:** Michael Bording-Jorgensen, Brendon Parsons, Jonas Szelewicki, Colin Lloyd, Linda Chui

**Affiliations:** 1Department of Laboratory Medicine and Pathology, University of Alberta, Edmonton, AB T6G 2R3, Canada; bordingj@ualberta.ca (M.B.-J.); BrendonP@dal.ca (B.P.); jszelewi@ualberta.ca (J.S.); cdlloyd@ualberta.ca (C.L.); 2Alberta Precision Laboratories-Public Health Laboratory (ProvLab), Edmonton, AB T6G 2J2, Canada

**Keywords:** STEC, serotyping, qPCR, broth 5, culture

## Abstract

Non-O157 Shiga toxin-producing *E. coli* (STEC) can cause outbreaks that have great economic and health impact. Since the implementation of STEC screening in Alberta in 2018, it is also essential to have a molecular serotyping method with faster turnaround time for cluster identification and surveillance purposes. This study sought to perform molecular serotyping of the top six non-O157 (O26, O45, O103, O111, O121 and O145) STEC serotypes directly from stools and enrichment broths compared to conventional methods on isolates. Multiplex, serotyping qPCR assays were used to determine sensitivity and specificity of the top six non-O157 STEC serotypes. Sensitivity and specificity were assessed for both singleplex and multiplex qPCR assays for comparison of the top six serotypes. Blinded stool specimens (*n* = 116) or broth samples (*n* = 482) submitted from frontline microbiology laboratories for STEC investigation were analyzed by qPCR. Both singleplex and multiplex assays were comparable, and we observed 100% specificity with a limit of detection of 100 colony-forming units per mL. Direct molecular serotyping from stool specimens mostly correlated (88%) with conventional serotyping of the cultured isolate. In cases of discordant serotypes, the top six non-O157 STEC mixed infections were identified and confirmed by culture and conventional serotyping. Detection of non-O157 STEC can be done directly from stool specimens using multiplex PCR assays with the ability to identify mixed infections, which would otherwise remain undetected by conventional serotyping of a single colony. This method can be easily implemented into a frontline diagnostic laboratory to enhance surveillance of non-O157 STEC, as more frontline microbiology laboratories move to culture independent assays.

## 1. Introduction

Shiga toxin-producing *Escherichia coli* (STEC) are one of the major causes of acute gastroenteritis (AGE) in Canada, which may result in complications and hospitalization [1]. *E. coli* O157, one of the most prominent STEC serotypes (synonymous with Verotoxigenic *Escherichia coli*, VTEC), has been implicated in various food-borne outbreaks world-wide [1]. AGE caused by STEC can lead to hemolytic uremic syndrome (HUS), a potentially fatal condition resulting in kidney failure. HUS is due to the presence of two Shiga toxin virulence factors, encoded *stx*_1_ and *stx*_2_ within the lambdoid prophages that inhibit protein synthesis [2]. The elderly and pediatric populations are most at risk of developing complications during infection. The Stx2 toxin has been shown to be more severe and associated with the development of HUS [3]. Furthermore, STEC require a very low infectious dose (fewer than 100 colony-forming units (CFU/mL)) and therefore have the potential to cause large numbers of outbreaks [4].

Ruminants, particularly cattle, have been identified as the primary reservoir for STEC where they do not cause the same disease but are rather shed into waterways or cause food-associated disease [5]. One of the most widely recognized cases of waterway contamination due to STEC was Walkerton, Ontario in 2000, where 2300 individuals were affected, causing 65 hospitalizations and 7 deaths [6]. Over 600 STEC serotypes have been identified (though not all are associated with human disease) from various sources, including the environment, food, humans and animals, with outbreak serotype predominance varying depending on the geographic region [7,8,9]. In 2011, *E. coli* O104:H4 was responsible for a major outbreak in Germany that resulted in 3816 cases, 845 HUS and 54 deaths over a period of 3 months [10], clearly demonstrating that STEC non-O157 infection can be catastrophic. In North America, the most commonly observed “top six” serotypes are O26, O45, O103, O111, O121 and O145 [11], which some have, on rare occasions, resulted in HUS (O26, O103, O111 and O121) [12].

Current diagnostic identification of O157 STEC infection relies on culture using MacConkey agar. We have recently shown that CHROMagar™ STEC agar plates were better able to identify non-O157 STEC isolates from stool samples as compared to the traditional Sorbitol–MacConkey agar used in STEC O157 detection [12]. We have also shown that non-O157 STEC isolates can be difficult to enrich in certain broth media, suggesting a potential for mixed infections that can be missed during an outbreak [13]. Culture-independent diagnostic tests (CIDT), which detect the presence of the *stx* gene(s), are becoming more readily available, allowing for rapid diagnosis, as well as increased surveillance [14,15]. CIDT are optimal for diagnostic identification, as they do not require culture prior to testing, which greatly decreases the turnaround time. However, conventional culture and STEC isolation are required for surveillance and serotype identification purposes using pulsed-field gel electrophoresis and whole genome sequencing [16,17]. In Alberta, STEC (both O157 and non-O157) screening began in 2019 alongside CIDT implementation in frontline microbiology laboratories. STEC identified by frontline laboratories were referred to Alberta Precision Laboratories ProvLab (APL-ProvLab) for isolation using chromogenic agar and colonies confirmed by PCR as part of the surveillance program. Since the implementation of this algorithm, over 60% of STEC detected were non-O157, causing diarrhea in this province. Non-O157 serotyping is performed at the Public Health Agency of the Canada National Microbiology Laboratory (PHAC-NML) located in Winnipeg, Manitoba, and thus the shipping of isolates for serotyping further increases the turnaround time for reporting. This study sought to assess the performance of a molecular serotyping qPCR assay directly from stools and enriched broth culture samples to predict the top six non-O157 (O26, O45, O103, O111, O121 and O145) serotypes as compared to the conventional isolate serotyping method.

## 2. Materials and Methods

### 2.1. Sensitivity Assay

One of each of the top six non-O157 STEC (O26, O45, O103, O111, O121 and O145) clinical human isolate stocks were retrieved from frozen skim milk stored at −80 °C and cultured on sheep blood agar plates (BAP) (Dalynn Biologicals, Calgary, AB, Canada). After overnight incubation at 37 °C, single colonies from each culture plate were then grown overnight at 37 °C in Tryptic Soy Broth (TSB) prepared by APL-ProvLab in Edmonton, Alberta, Canada. Each enriched culture broth was diluted to an optical density of 0.5 and performed ten-fold serial dilutions from 10^−1^ to 10^−8^. An aliquot of 100 μL was inoculated onto BAP (Dalynn Biologicals, Calgary, Canada) in triplicates and incubated overnight to determine the CFU/mL. Aliquots of cell suspensions containing 10^2^–10^4^ CFU/mL were centrifuged at 13,000× *g* for 5 min; supernatant was discarded, and DNA was extracted from the pellet by boiling in a rapid lysis buffer (100 mM NaCl, 10 mM Tris-HCL pH 8.3, 1 mM EDTA pH 9.0, 1% Triton X-100) as described previously [18]. An amount of 5 μL DNA was used as template in a total of 25 μL reaction volume for serotyping as singleplex or multiplex assays, as shown in Table 1. The amplification contained 12.5 μL of 1× PrimeTime^®^ Gene Expression Master Mix (Integrated DNA Technology, Skokie, IL, USA), 0.33 μM of each primer, 0.22 μM probe, 5 μL DNA template and molecular biology grade water in a total of 25 μL reaction volume. A positive and no-template controls were included in each run, and the qPCR conditions consisted of 95 °C for 1 min followed by 40 cycles of 95 °C for 5 s and 58 °C for 45 s, performed on the 7500 FAST qPCR system (Applied Biosystems, Foster City, CA, USA).

### 2.2. Specificity Assay

Specificity was determined using a panel of non-STEC isolates consisting of the following isolates; *Aeromonas hydrophila*, *Salmonella* serovar Enteriditis, *Salmonella* serovar Typhimurium, *Shigella boydii* (Serotype 2), *Shigella flexneri* (Serotype 2), *Shigella sonnei, Shigella dysenteriae* (Serotype 2), *Yersinia enterocolitica*; and 13 reference strains consisted of *Proteus mirabalis* (ATCC12453), *Campylobacter coli* (ATCC33559), *Campylobacter jejuni* (ATCC33291), *Citrobacter freundii* (ATCC8090), *Klebsiella pneumoniae* (ATCC31488), *Enterobacter cloacae* (ATCC13047), *Staphylococcus aureus* (ATCC25923), *Staphylococcus aureus* (ATCC25913), *Escherichia coli* (ATCC25922), *Bacillus cereus* (ATCC14579) and *Staphylococcus epidermis* (ATCC12228). As for inclusivity panel, the following clinical STEC strains (*n* = 201) were included. These were comprised of O26 (*n* = 34), O45 (*n* = 3), O103 (*n* = 39), O111 (*n* = 35), O121 (*n* = 29), O145 (*n* = 17) in addition to non-top six isolates (*n* = 44), as shown in Appendix A.

Each isolate was grown on BAP (Dalynn Biologicals, Calgary, Canada) under the respective growth conditions; single-colony DNA extraction was performed using a single colony suspended in rapid lysis and performed the procedure as described in previous section. PCR assays were performed as singleplex and multiplex assays with both sets of primers and probes, as described above.

### 2.3. Patient Samples

All patient samples were either STEC-positive stool samples and/or enriched cultures from positive stool using either Gram Negative broth (GNB) (Dalynn Biologicals, Calgary, Canada) provided by Calgary Laboratory Services or Tryptic Soy Broth (TSB) (Dalynn Biologicals, Calgary, Canada) from DynaLIFE (Edmonton, AB, Canada). These samples were previously screened positive by SHIGA TOXIN QUIK CHEK™ immunoassay (TechLab Inc, Blacksburg, VA, USA) by the providing laboratories as non-O157 STEC. Workflow for all clinical samples is shown in Figure 1, with specific methods in the following Section 2.4 and Section 2.5.

### 2.4. Direct Molecular Serotyping from Clinical Samples

Enriched culture in TSB and GNB (*n* = 482) were extracted using rapid lysis buffer. An amount of 250 µL of enriched culture was centrifuged (13,000× *g* for 10 min), supernatant was discarded, and the cell pellet was resuspended in rapid lysis buffer and heated to 95 °C for 15 min using a heating block (Thermo Fisher, Mississauga, Canada). The samples were then briefly centrifuged (13,000× *g* for 10 min), and the supernatant stored at 4 °C until qPCR was performed. Cycle threshold above 0.1 Delta Rn and Ct value ≤ 36 were considered positive.

DNA from stool samples (*n* = 116) were extracted using the NucliSENS easyMag system (bioMerieux, Montreal, QC, Canada). Approximately 10 µL loop of stool was suspended in 1 mL of NucliSENS Lysis buffer in SK38 soil grinding lysis bead tubes (Luminex Corporation, Toronto, ON, Canada) and shaken on a vortex at max speed for 10 min; a sample was left at room temperature for 15 min and followed by centrifugation (15,871× *g* for 5 min). A total of 200 µL was extracted and with a final elution volume of 70 µL.

The primers’ and probes’ (Integrated DNA Technology, Coralville, IA, USA) design was based on reference sequences [19], as shown in Table 1. The total reaction contained 12.5 μL of 1× PrimeTime^®^ Gene Expression Master Mix (Integrated DNA Technology, Coralville, USA), 0.33 μM of each primer, 0.22 μM probe, 5 μL DNA template and molecular biology grade water in a total of 25 μL reaction volume. A no-template control was included in each run. qPCR amplification conditions consisted of 95 °C for 1 min, followed by 40 cycles of 95 °C for 5 s and 58 °C for 45 s performed on the 7500 FAST qPCR system (Applied Biosystems, Mississauga, Canada).

### 2.5. Isolation of Non-O157 STEC

All enriched broth cultures (100 μL) originally from positive STEC clinical stool samples were inoculated onto CHROMagar™ STEC plates (Micronostyx, Ottawa, Canada) for strain isolation and further characterization. Individual mauve color colonies (*n* = 3) of the same morphotype were picked from each plate and suspended in rapid lysis buffer, heated, centrifuged, and 5 μL of the supernatant was used for STEC confirmation using our in-house qPCR. Once confirmed as STEC by *stx* identification, the isolate was forwarded to PHAC-NML for conventional serotyping. Serotyping results from stools, enriched cultures and isolates were compared, and if discordance was observed, then the samples were considered to contain >1 serotypes indicative of a mixed infection. Additional isolates were picked for qPCR serotyping to identify the remaining serotype and were subsequently sent to the PHAC-NML for confirmation (Figure 1).

### 2.6. Statistics

Sensitivity and specificity data were analyzed using Excel (Microsoft Office). Comparison between singleplex and multiplex was analyzed with a Wilcoxon matched pairs signed-rank test using Prism9 for Mac (Graph Pad, San Diego, CA, USA).

## 3. Results

### 3.1. Sensitivity and Specificity

The singleplex and multiplex assays using primers and probes originating from Perelle et al. [19] were found to be highly specific and sensitive, with a detection limit of 1 × 10^2^ CFU/mL, as shown in Table 2. The results between the singleplex and multiplex of both assays do not show major differences (*p* > 0.05), as illustrated by the crossing point of the qPCR (Table 2). As for the specificity assay, no cross reactivity was observed between the isolates of the non-top six STEC or the exclusivity panel of different bacterial species. We observed 100% specificity, as well as a true positive rate of 100%, based on the 157 top six isolates, for the qPCR assays.

### 3.2. Clinical Samples Testing

We were able to determine the serotype from stool samples (*n* = 116) as well as enriched broths (*n* = 482) from stools using the direct molecular serotyping method. Isolates obtained from the enrichment broths were sent to the PHAC-NML for conventional serotyping and were compared to the direct-from-stool/broth serotyping assay. Concordance between direct serotyping from stool, broth and conventional isolate serotyping were found for 102 samples (88%) (Table 3); 1 sample was found to have discordance between direct serotyping from stool and broth to conventional serotyping (Table 3). The majority of the discordance was found to be between direct serotyping from stool (stool-only column) to direct serotyping from broth and conventional isolate serotyping, with one stool being positive for two top-six serotypes (*n* = 13 samples, 11.2%) (Table 3). Concordance was also measured with the remaining broth samples (*n* = 366 samples), which we did not receive a matching stool sample for (Table 4). We observed a 97.5% concordance between broths and isolates. There were eight serotypes that were identified through the enriched broths, and other serotypes were found when the isolates were picked from agar plates for traditional serotyping. The STEC isolates that failed to be typed by the top six molecular serotypes were classified as non-top six, and their serotypes were determined by the PHAC-NML, as shown in Appendix A. Next, we compared the *stx* characteristics between the isolates and found most of the infections were from *stx*_1_ (Table 5).

### 3.3. Mixed Infections between Broth and Stool Samples

Any discordant sample between direct-from-stool/broth qPCR and conventional serotyping were treated as if they were mixed infections and were further investigated. For discordant serotyping, results from one stool and eight additional broth samples, which were initially typed as top six, but the picked colony determined by conventional typing gave discordant results, further colonies from the CHROMagar™ STEC plates were subjected to qPCR to confirm as mixed infection. Colonies from six of these samples were subtyped by conventional serotyping, and the confirmed mixed infections are shown in Table 6. For the remaining three samples, we failed to isolate any additional colonies, although the broths tested positive by PCR. In addition, there were 13 stool samples with discordant results as compared to broth and conventional serotyping, which we were not able to isolate. There was no pattern with regard to serotypes involved in mixed infections, and most were discovered when the in-house qPCR detected a top-six serotype, but the PHAC-NML result from the single isolate was reported as a non-top-six serotype (Appendix A).

## 4. Discussion

STEC remain one of the most prevalent food-borne associated infectious diseases in Canada, with Alberta having the highest burden/prevalence of the disease [20]. Currently, little data have been published on the incidence of non-O157 infections within Alberta prior to the implementation of routine STEC screening in 2018. Although the O157 serotype is the most recognized cause of *E. coli*-induced gastroenteritis, non-O157 outbreaks have been increasing in recent years. According to the EU One Health Zoonosis report in 2020, there was an overall trend for reporting STEC cases between 2015 to 2019 due to the awareness of the importance of STEC detection associated with outbreaks in the EU [21]. In Canada, there was a recommendation in 2018 from the Canadian Public Health Laboratories Network for all Canadian laboratories to detect all STEC, especially the non-O157, for better surveillance purposes [14]. An outbreak of O121 in 2016 linked to wheat flour was confirmed using pulsed-field gel electrophoresis and whole genome sequencing [22], of which both techniques require growth of the pathogen. Culture is essential for cluster detection and surveillance; however, direct stool serotyping will give the laboratory an early indication of the top six non-O157 STEC serotypes along with the *stx* gene status. Culture is still required after this initial step, but it will shorten the turnaround time for reporting. Our direct stool serotyping method is only limited to the top six serotypes, which consisted of the majority (72.4%) of samples tested, but any non-top-six samples still require conventional serotyping method at the PHAC-NML.

Sensitivity and specificity were comparable between both multiplex and singleplex assays, indicating that the multiplex assay is sufficient for implementation to support existing STEC surveillance programs. The primers also showed high specificity, indicating they could distinguish between the different serotypes, which is crucial in cases of mixed infections. When clinical samples were tested, O26 was the predominant serotype identified, followed by O111. Surprisingly, there were multiple non-top six STEC involved in diarrheal disease found in both stool and enriched broth samples. As for mixed infections, all the confirmed samples contained both top-six and non-top-six serotypes and were found through discordance between serotyping methods. This suggests that our current approach likely underestimates the number of non-top-six serotypes involved in infections and outbreaks.

One of the limitations of this study is the fact that we only analyzed the top six serotypes, and it is possible there are mixed infections with less common serotypes of STEC. However, it would not be economical for diagnostic laboratories to screen every sample for hundreds of potential mixed infections with very rare serotypes. As only a single colony was picked from a plate to be sent for conventional serotyping, it is possible that non-top-six STEC can be missed; therefore, more colonies could be picked and conventionally serotyped to increase surveillance of the non-top-six STEC. Although we determined the limit of detection was 100 cells using pure culture, it is still possible that the bacterial load in a stool sample is below the detectable limit, and therefore a serotype may false-negatively be assumed to be non-top six. We observed seven stool samples where the top-six serotype was not detectable until after enrichment in TSB broth, which is a limitation of relying solely on stool samples for any diagnostic test. However, enrichment into broth introduces the possibility that mixed infections may be missed due to the growth of the more abundant serotype. Another possibility of missing serotypes is the fact that stool samples are incredibly complex and may contain inhibitors that interfere with qPCR, particularly for those serotypes that have a low abundance, which is likely why we had a higher number of discordances with stool samples (Appendix A). Interestingly, this seems to have been the case with O103 stool samples (five samples), as most of the mismatched isolates (only detected by one of the serotyping methods) were due to this serotype. This may indicate O103 primers as being less sensitive, particularly when dealing with stool samples, as they also had a higher Ct value for pure culture as compared to the other serotypes (Appendix A).

In conclusion, we have shown that direct-from-stool serotyping can be performed for positive STEC stools using multiplex qPCR. Furthermore, we have identified the potential for mixed infections to occur. This assay can be implemented alongside routine diagnostic tools for rapid identification of the top six serotypes that cause the majority of STEC outbreaks in North America.

## Figures and Tables

**Figure 1 microorganisms-10-00329-f001:**
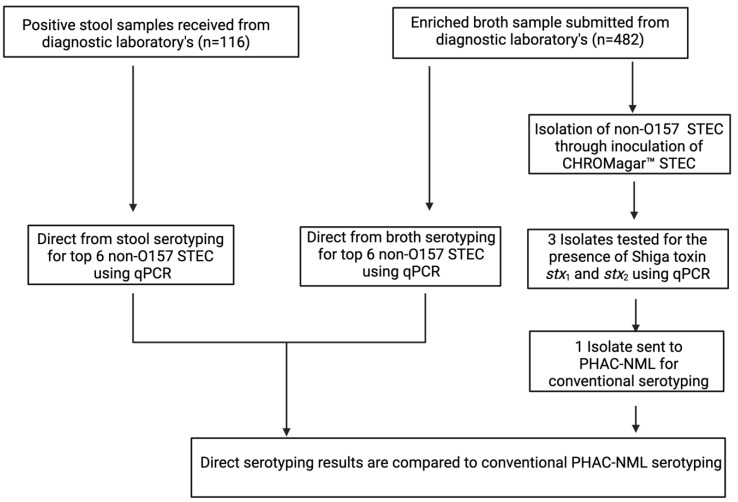
STEC serotyping workflow comparing enriched broth culture and stool direct serotyping to conventional serotyping provided by the Public Health Agency of Canada—National Microbiology Laboratory (PHAC-NML), Winnipeg, Manitoba, Canada. Positive stool (*n* = 116) and broth (*n* = 482) samples were directly molecularly serotyped using qPCR. Additionally, broth cultures were plated onto CHROMagar™ STEC (Micronostyx. Ottawa, ON, Canada) where 3 colonies were picked and characterized for *stx* using qPCR. One of the isolates was then sent to the PHAC-NML for conventional serotyping and result generated was then compared to the in-house direct serotyping. Any discordant result between direct and conventional serotyping was treated as a mixed infection and an attempt was made to isolate the additional serotype for confirmation.

**Table 1 microorganisms-10-00329-t001:** Primer and probe sequences used in this study.

Reference Gene, Primer/Probe	Sequence 5′–3′ [19]
*stx*_1_-F	TTT GTY ACT GTS ACA GCW GAA GCY TTA CG
*stx*_1_-R	CCC CAG TTC ARW GTR AGR TCM ACR TC
*stx*_1_-P	CTG GAT GAT CTC AGT GGG CGT TCT TAT GTA A
*stx*_2_-F	TTT GTY ACT GTS ACA GCW GAA GCY TTA CG
*stx*_2_-R	CCC CAG TTC ARW GTR AGR TCM ACR TC
*stx*_2_-P	TCG TCA GGC ACT GTC TGA AAC TGC TCC
*wzx* O26-F	CGC GAC GGC AGA GAA AAT T
*wzx* O26-R	AGC AGG CTT TTA TAT TCT CCA ACT TT
*wzx* O26-P	CCC CGT TAA ATC AAT ACT ATT TCA CGA GGT TGA
*wzx* O45-F	TAC GTC TGG CTG CAG GG
*wzx* O45-R	ACT TGC AGC AAA AAA TCC CC
*wzx* O45-P	TTC GTT GCG TTG TGC ATG GTG GC
*wzx* O103-F	CAA GGT GAT TAC GAA AAT GCA TGT
*wzx* O103-R	GAA AAA AGC ACC CCC GTA CTT AT
*wzx* O103-P	CAT AGC CTG TTG TTT TAT
*wbdl* O111-F	CGA GGC AAC ACA TTA TAT AGT GCT TT
*wbdl* O111-R	TTT TTG AAT AGT TAT GAA CAT CTT GTT TAG C
*wbdl* O111-P	TTG AAT CTC CCA GAT GAT CAA CAT CGT GAA
*wzx* O121-F	TGG TCT CTT AGA CTT AGG GC
*wzx* O121-R	TTA GCA ATT TTC TGT AGT CCA GC
*wzx* O121-P	TCC AAC AAT TGG TCG TGA AAC AGC TCG
*ihpl* O145-F	CGA TAA TAT TTA CCC CAC CAG TAC AG
*ihpl* O145-R	CCG CCA TTC AGA ATG CAC ACA ATA TCG
*ihpl* O145-P	ACA GTG CCA GCA TTC GCT TGC GA

In the sequences: Y is (C, T), S is (C, G), W is (A, T), R is (A, G), M is (A, C).

**Table 2 microorganisms-10-00329-t002:** Singleplex and Multiplex qPCR of Non-O157 STEC Isolates.

**Singleplex (Ct Values)**						
**Primer/Probe Set**	**O26**	**O111**	**O45**	**O121**	**O103**	**O145**
Target Organism	*E. coli* O26	*E. coli* O111	*E. coli* O45	*E. coli* O121	*E. coli* O103	*E. coli* O145
10^4^ (CFU/mL)	25.6	27.5	26.3	25.2	29.9	25.5
10^3^ (CFU/mL)	29.3	30.8	30.6	28.8	33.5	28.7
10^2^ (CFU/mL)	35	34.3	33	33.8	35	31.8
**Multiplex (Ct Values)**						
**Primer/Probe Set**	**O26/O45**	**O26/O45**	**O121/O145**	**O121/O145**	**O103/O111**	**O111/O103**
Target Organism	*E. coli* O26	*E. coli* O45	*E. coli* O145	*E. coli* O121	*E. coli* O103	*E. coli* O111
10^4^ (CFU/mL)	25.6	26.3	25.4	25.3	29.4	27.4
10^3^ (CFU/mL)	29.4	30.5	28.9	28.9	33.4	31.3
10^2^ (CFU/mL)	35	32	31.5	33	35	34.6

**Table 3 microorganisms-10-00329-t003:** Serotypes identified through direct stool or broth serotyping and conventional isolate serotyping (*n* = 116) *.

Serotype	Stools/Broths/Isolate	Stools/Broths	Stools/Isolate	Broths/Isolate	Broth-Only	Isolate-Only	Stool-Only
*E. coli* O26	37	1	0	1	0	0	2 ^#^
*E. coli* O111	17	0	0	0	0	0	1 ^#^
*E. coli* O121	7	0	0	1	0	0	3
*E. coli* O103	10	0	0	8	0	0	1
*E. coli* O145	6	0	0	0	0	0	0
*E. coli* non-top six	25	0	0	3	0	1	7

* Broth-Only, Isolate-Only and Stool-Only columns are discordant results that may indicate mixed infections. ^#^ Indicates stool sample in which two top five serotypes were detected.

**Table 4 microorganisms-10-00329-t004:** Serotypes identified through direct broth serotyping and conventional isolate serotyping (*n* = 366) *.

Serotype	Broths/Isolate	Broth-Only	Isolate Only
*E. coli* O26	106	1	0
*E. coli* O111	63	1	1
*E. coli* O121	25	0	1
*E. coli* O103	50	4	0
*E. coli* O145	11	1	0
*E. coli* non-top six	102	2	7

* Broth-Only and Isolate-Only columns are discordant results that may indicate mixed infections.

**Table 5 microorganisms-10-00329-t005:** *stx* Characteristics of the isolates (*n* = 488).

Serotype	*stx_1_*	*stx* _2_	*stx*_1_ and *stx*_2_
*E. coli* O26	117	13	16
*E. coli* O111	64	10	7
*E. coli* O121	31	4	0
*E. coli* O103	64	2	4
*E. coli* O145	12	4	2
*E. coli* non-top six	105	26	7

**Table 6 microorganisms-10-00329-t006:** Mixed infections identified using in-house multiplex qPCR.

In-House Direct Serotype	Conventional Serotype Based on Picked Colonies from CHROMagar™ STEC Plate
O26	O69
O145	O27
O111	O: undetermined
O26	O: undetermined
O103	O187
O103	O69

## Data Availability

Not applicable.

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
