# Peer review of "Molecular Detection of Non-O157 Shiga Toxin-Producing *Escherichia coli* (STEC) Directly from Stool Using Multiplex qPCR Assays"

_microorganisms, 2022, doi:10.3390/microorganisms10020329_

Round 1

Reviewer 1 Report

Any discordant sample between direct from stool/broth qPCR and conventional serotyping cannot be considered as a consequence of a mixed infection, since there are strains of serogroups O26, O45, O103, O111, O121 and O145 that are not STEC.

In this study, the existence of mixed infections caused by STEC strains belonging to different serogroups has not been demonstrated. To demonstrate the existence of mixed infections, it is necessary to carry out the culture and isolation of the STEC strains belonging to two different serogroups.

Author Response

Reviewer 1:

Any discordant sample between direct from stool/broth qPCR and conventional serotyping cannot be considered as a consequence of a mixed infection, since there are strains of serogroups O26, O45, O103, O111, O121 and O145 that are not STEC.

In this study, the existence of mixed infections caused by STEC strains belonging to different serogroups has not been demonstrated. To demonstrate the existence of mixed infections, it is necessary to carry out the culture and isolation of the STEC strains belonging to two different serogroups.

Response 1: We thank the reviewer for the comment and apologize for not clarifying this point more clearly in the manuscript. We agree that confirmation of mixed infections requires culture and isolation, which we successfully did with 6 patient samples (Table 6). These isolates were picked from culture media, confirmed as STEC by PCR before conducting conventional serotyping. Unfortunately, STEC can be difficult to isolate from samples with a low bacterial load and therefore it is likely that we might have missed some of the mixed infections with non-top 6 STEC.

Reviewer 2 Report

This is an interesting and useful study that has attempted to establish an efficient molecular serotyping of the top 6 non-O157 STEC serotypes directly from stools and enrichment broths and to compare the molecular serotyping with conventional serotyping methods. The authors have performed extensive molecular analysis in a significant number of samples and this is one of the strong points of the study. However, the way that the results are presented may be confusing in certain areas and better explanations are needed for the reader to fully comprehend the findings.

Some remarks:

Line 84: Was milk pasteurized or not? Was it directly from farms or from the market? From how many different samples were the isolates retrieved?

Line 112: Remove comma before STEC strains

Line 104: Where these isolates from milk also? Please clarify. How were they identified?

Line 112-113: These STEC strains were the ones isolated from milk?

Line 122: In Figure 1 please explain PHAC-NML

Line 130: It is not clear from what kind of clinical samples were the enriched cultures (stools, urine?) Were they from isolates of the 116 stool samples?

Line 158: “All positive enriched cultures (100 μL) from clinical stool samples were inoculated onto CHROMagar™”, please clarify:

  • Weren’t all enriched cultures positive? If yes then please rephrase because it is confusing as this phrase implies that some were not positive. However, in Line 133 it states that samples were screened as positive.
  • If enriched cultures were also from clinical stool samples, please also clarify earlier in text

Line 165: Stool samples were also sent for conventional serotyping? Were they also inoculated in Chomagar?

Line 174: In this section, an initial phrase to explain the results that are following would be helpful to connect with the materials and methods, eg  The results of the Non-O157 STEC (n=202) isolates from milk have shown…

Table 2: The Perelle et al. citation would be better placed in the legend of table. Also, it would be better if statistical analysis was shown to demonstrate that there were no statistically significant differences

Table 3: From this table the reader can finally understand that the authors have treated the samples as “Stools/ Broths/ Isolate” units, something that was not clear at the Materials and Methods section. Please provide more clear explanations at the text and at Figure 1. Moreover, regarding the columns Stools/ Broths, Stools/ Isolate, Broths/ Isolate, aren’t they also discordant results?  Should they be added in the note “* Broth Only, Isolate Only, and Stool Only columns are discordant results that may indicate mixed infections”?

In the Discussion, please consider adding as citation the latest report from EFSA (2020) in which there are key statistics on STEC strains in EU, indicating the globality of the problem:

https://efsa.onlinelibrary.wiley.com/doi/epdf/10.2903/j.efsa.2021.6971

Also, in the Discussion, the fact that you have retrieved 202 STEC isolates from a common food source, is a fact that deserves mentioning to emphasize that STEC strains can be an important public health issue.

Supplemental Table 2: In the title: “Discordant Serotypes Between Stool Samples and Conventional” is it not “between stool molecular serotype and conventional serotype”?

Author Response

Reviewer 2:

This is an interesting and useful study that has attempted to establish an efficient molecular serotyping of the top 6 non-O157 STEC serotypes directly from stools and enrichment broths and to compare the molecular serotyping with conventional serotyping methods. The authors have performed extensive molecular analysis in a significant number of samples and this is one of the strong points of the study. However, the way that the results are presented may be confusing in certain areas and better explanations are needed for the reader to fully comprehend the findings.

Some remarks:

  1. Line 84: Was milk pasteurized or not? Was it directly from farms or from the market? From how many different samples were the isolates retrieved?

Response 1: We apologize for the misunderstanding regarding the skim milk isolates. Skim milk is using as a preservative for long term storage at -80 ºC and not source isolation. We have modified the sentence to “One of each of the top 6 non-O157 STEC (O26, O111, O45, O121, O103 and O145) clinical human isolates”. We have also included a supplemental table indicating which non-top 6 serotypes were used.

  1. Line 112: Remove comma before STEC strains

Response 2: We thank the reviewer for the suggestion, and we have removed the comma.

  1. Line 104: Where these isolates from milk also? Please clarify. How were they identified?

Response 3: Again, we apologize for this misunderstanding. These are previously characterized human clinical isolates stored in skim milk to use as the sensitivity panel. Skim milk isolates were used for the sensitivity only. Clinical samples are those from either stool or enriched broth.

  1. Line 122: In Figure 1 please explain PHAC-NML

Response 4: PHAC-NML is the Public Health Agency of Canada-National Microbiology Laboratory as shown in the methods. We have added this to the figure legend for clarification.

  1. Line 130: It is not clear from what kind of clinical samples were the enriched cultures (stools, urine?) Were they from isolates of the 116 stool samples?

Response 5: We apologize for this misunderstanding. The enriched broth culture was all from stool samples. The stool and/or broth samples were received from the frontline microbiology diagnostic laboratory. Of the 482 broth samples (cultured from stool), we were also given 116 corresponding stool samples. The remaining broth samples, we were not able to get the original stool. This has been clarified in the methods section line 117 “All patient samples were either STEC positive stool samples and/or enriched cultures from stool using”.

  1. Line 158: “All positive enriched cultures (100 μL) from clinical stool samples were inoculated onto CHROMagar™”, please clarify:
  • Weren’t all enriched cultures positive? If yes then please rephrase because it is confusing as this phrase implies that some were not positive. However, in Line 133 it states that samples were screened as positive.
  • If enriched cultures were also from clinical stool samples, please also clarify earlier in text

Response 6: We thank the reviewer for the comment and apologize for the misunderstanding. All of the enriched broth cultures were screened by the submitting diagnostic lab. We have changed line 148 to be “All enriched broth cultures (100 μL) originally from STEC positive clinical stool samples were inoculated onto CHROMagar™ STEC plates”.

  1. Line 165: Stool samples were also sent for conventional serotyping? Were they also inoculated in Chomagar?

Response 7: We thank the reviewer for the comment. Stool samples were not inoculated onto CHROMagar plates as this arm was to determine whether serotyping could be done directly from the stool. Furthermore, all of the stool samples had a corresponding broth sample which was inoculated onto CHROMagar for isolate identification.

  1. Line 174: In this section, an initial phrase to explain the results that are following would be helpful to connect with the materials and methods, eg  The results of the Non-O157 STEC (n=202) isolates from milk have shown…

Response 8: We thank the reviewer for the suggestion. We have clarified in the methods section 2.5 line 148 as well as results section 3.2 line 171 to clarify the enriched broth is from a clinical stool sample.

  1. Table 2: The Perelle et al. citation would be better placed in the legend of table. Also, it would be better if statistical analysis was shown to demonstrate that there were no statistically significant differences

Response 9: We thank the reviewer for the suggestion and have removed the Perelle et al from the table as the primers are already referenced in Table 1. As for statistical significance, we performed a Wilcoxon matched-pairs signed rank test and found no significant difference and indicated on line 165-166.

  1. Table 3: From this table the reader can finally understand that the authors have treated the samples as “Stools/ Broths/ Isolate” units, something that was not clear at the Materials and Methods section. Please provide more clear explanations at the text and at Figure 1. Moreover, regarding the columns Stools/ Broths, Stools/ Isolate, Broths/ Isolate, aren’t they also discordant results?  Should they be added in the note “* Broth Only, Isolate Only, and Stool Only columns are discordant results that may indicate mixed infections”?

Response 10: We thank the reviewer for the suggestion and apologize for the misunderstanding regarding the results and workflow. We have added more information in the methods section regarding sample workflow and analysis. We considered discordant as those serotypes identified by only one source (stool, broth, or isolate) whereas concordant results are those with serotype identified from 2 or more sources.

  1. In the Discussion, please consider adding as citation the latest report from EFSA (2020) in which there are key statistics on STEC strains in EU, indicating the globality of the problem:

https://efsa.onlinelibrary.wiley.com/doi/epdf/10.2903/j.efsa.2021.6971

Response 11: We thank the reviewer for suggesting the addition of this reference and have added a sentence to our discussion.

  1. Also, in the Discussion, the fact that you have retrieved 202 STEC isolates from a common food source, is a fact that deserves mentioning to emphasize that STEC strains can be an important public health issue.

Response 12: We thank the reviewer for the comment. We agree that STEC is an important public health issue and can be isolated from many common food sources as highlighted in the references regarding recent outbreaks associated with non-O157 STEC. However, as previously stated the 202 STEC were not from milk but were stored in skim milk as a cryopreservative.

  1. Supplemental Table 2: In the title: “Discordant Serotypes Between Stool Samples and Conventional” is it not “between stool molecular serotype and conventional serotype”?

Response 13: We thank the reviewer for the comment and agree with their suggestion and have changed the title of the supplemental table.

Round 2

Reviewer 2 Report

The revised manuscript is accepted